Compositional and predicted functional analysis of the gut microbiota of Radix auricularia (Linnaeus) via high-throughput Illumina sequencing

Hu Zongfu 1 2 3
Chen Xi 2
Chang Jie 1
Yu Jianhua 1
http://orcid.org/0000-0002-6733-7340 Tong Qing 2
Li Shuguo 1
Niu Huaxin 1 niuhx@imun.edu.cn
1 College of Animal Science and Technology, Inner Mongolia University for Nationalities , Tongliao , People’s Republic of China
2 College of Animal Science and Technology, Northeast Agricultural University , Harbin , People’s Republic of China
3 Inner Mongolia Key Laboratory of Toxicant Monitoring and Toxicology , Tongliao , People’s Republic of China
LaMontagne Michael
Electronic publication date: 2018 Aug 28
Publication date: 2018
Volume: 6
Electronic Location ID: e5537
Received 2018 Feb 12; Accepted 2018 Aug 8
Copyright: © 2018 Hu et al.
Copyright year: 2018
Copyright holder: Hu et al.
License: This is an open access article distributed under the terms of the Creative Commons Attribution License, which permits unrestricted use, distribution, reproduction and adaptation in any medium and for any purpose provided that it is properly attributed. For attribution, the original author(s), title, publication source (PeerJ) and either DOI or URL of the article must be cited.
License URL: https://creativecommons.org/licenses/by/4.0/

Keywords: Radix auricularia, Intestinal bacterial communities, 16S rRNA gene, Illumina Miseq sequencing

Funding: National Natural Science Foundation of China 31360640 and 314360692 Natural Science Foundation of Inner Mongolia University for Nationalities NMDYB1705 Interdisciplinary Program for Inner Mongolia University for Nationalities MDXK008 This work was supported by the National Natural Science Foundation of China (No. 31360640 and 314360692), Natural Science Foundation of Inner Mongolia University for Nationalities (No. NMDYB1705), and Interdisciplinary Program for Inner Mongolia University for Nationalities (MDXK008). The funders had no role in study design, data collection and analysis, decision to publish, or preparation of the manuscript.

==============================
Due to its wide distribution across the world, the snail Radix auricularia plays a central role in the transferal of energy and biomass by consuming plant biomass in freshwater systems. The gut microbiota are involved in the nutrition, digestion, immunity, and development of snails, particularly for cellulolytic bacteria, which greatly contribute to the digestion of plant fiber. For the first time, this study characterized the gut bacterial communities of R. auricularia, as well as predicted functions, using the Illumina Miseq platform to sequence 16S rRNA amplicons. Both juvenile snails (JS) and adult snails (AS) were sampled. The obtained 251,072 sequences were rarefied to 214,584 sequences and clustered into 1,196 operational taxonomic units (OTUs) with 97% sequence identity. The predominant phyla were Proteobacteria (JS: 36.0%, AS: 31.6%) and Cyanobacteria (JS: 16.3%, AS: 19.5%), followed by Chloroflexi (JS: 9.7%, AS: 13.1%), Firmicutes (JS: 14.4%, AS: 6.7%), Actinobacteria (JS: 8.2%, AS: 12.6%), and Tenericutes (JS: 7.3%, AS: 6.2%). The phylum Cyanobacteria may have originated from the plant diet instead of the gut microbiome. A total of 52 bacterial families and 55 genera were found with >1% abundance in at least one sample. A large number of species could not be successfully identified, which could indicate the detection of novel ribotypes or result from insufficient availability of snail microbiome data. The core microbiome consisted of 469 OTUs, representing 88.4% of all sequences. Furthermore, the predicted function of bacterial community of R. auricularia performed by Phylogenetic Investigation of Communities by Reconstruction of Unobserved States suggests that functions related to metabolism and environmental information processing were enriched. The abundance of carbohydrate suggests a strong capability of the gut microbiome to digest lignin. Our results indicate an abundance of bacteria in both JS and AS, and thus the bacteria in R. auricularia gut form a promising source for novel enzymes, such as cellulolytic enzymes, that may be useful for biofuel production. Furthermore, searching for xenobiotic biodegradation bacteria may be a further important application of these snails.

Introduction

Radix auricularia (Linnaeus, 1758), a pulmonate snail, is naturally distributed in freshwater systems across both Europe and Asia (Stift et al., 2004; Vasileva, 2012). As a primary consumer, snails are common in freshwater systems, and both their energy and biomass can be transferred to fish, turtles, water birds, and mammals (Dewitt, Sih & Hucko, 1999; Eckblad, 1976). In addition to their role in the ecosystem, R. auricularia are intermediate hosts for many parasites (e.g., flukes), which are harmful to cattle, birds, and humans (Soldánová et al., 2010; Bargues et al., 2001).

The gut bacteria of snails or other animals are involved in multiple physiological processes of their hosts, primarily including digestion, nutrition, development, reproduction, immunity, and environmental resistance (Nicolai et al., 2015; Aronson, Zellmer & Goffredi, 2016; Sommer & Bäckhed, 2013; Pinheiro et al., 2015; Nayak, 2010). The capacity of decomposing lignocellulosic or pectic biomass increases their ability to utilize a variety of plant biomass, such as algae, water weeds, and leaf litters (Schamp, Horsák & Hájek, 2010; Vasileva, 2012). Beyond their own digestive enzymes, snails also utilize vast amounts of additional enzymes, secreted by bacterial activity within their gut, which assists in the digestion of up to 60–80% of the consumed plant fiber (Charrier et al., 2006). Cellulase secreted by snail gut bacteria is also used for industrial processes, including the production of biofuels from plant feedstocks (Cardoso et al., 2012a; Pinheiro et al., 2015; Pawar et al., 2015). Their use is more economic and eco-friendly than the use of acid hydrolysis and thermochemical methods (Hamelinck, Van Hooijdonk & Faaij, 2005). However, the currently available knowledge of the bacterial communities of the snail gut is limited.

Many factors can influence the composition of the bacterial community of snail and other animals, and previous studies have shown that prominent factors include the diet, season, pathogens, and physiological diseases (Cardoso et al., 2012b; Nicolai et al., 2015; Pawar et al., 2012; Stephens et al., 2016; Chandler et al., 2014). Furthermore, as reported for other animals (zebra fish, bovine, Atlantic salmon), gut microbiomes are also influenced by host development and growth stage (Jami et al., 2013; Nistal et al., 2012; Stephens et al., 2016; Llewellyn et al., 2016). A typical example is the Atlantic salmon (Salmo salar): its gut bacterial communities are more strongly influenced by life-cycle stage than by geography (Llewellyn et al., 2016).

Gut microbiota are also widely associated with reproductive processes. In the snail Potamopyrgus antipodarum, significant differences were found in the bacterial community composition between sexual and asexual snails, suggesting that reproductive mode influences microbiome composition, which way this relationship goes is still unclear (Takacs-Vesbach et al., 2016). As reported by a series of studies, gut bacteria are strongly involved in gonad development and reproduction, chiefly in improving spermatogenesis, oocyte maturation, and fecundity (Gioacchini et al., 2011; Carnevali, Maradonna & Gioacchini, 2017). Therefore, one aim of this study is to understand if the development or growth affects gut microbial communities as well as their functions.

In summary, gut bacteria in snails are relevant for physiological processes, the ecosystem, and industrial processes. Consequently, their diversity and function are worth exploring. Although few studies focused on the gut bacterial community of terrestrial snails (Cardoso et al., 2012a; Pawar et al., 2012; Nicolai et al., 2015; Charrier et al., 2006), studies of freshwater snails are rare. Here, we characterized the whole profile of the gut bacterial community of R. auricularia at different growth stages (juvenile and adult stage) and explored the roles of the obtained gut bacteria in other systems of the snail or the environment via functional prediction of bacteria metagenomic data. The results of this study fill an important gap in our knowledge of mollusks and provide important hints about their potential for technological applications and ecologic significance.

Materials and Methods

Research permits were provided by the Forestry Bureau of Tongliao (TL218) and by the Inner Mongolia University for Nationalities’ Institutional Animal Use and Care Committee (2016-IMUN-029).

Sample collection

R. auricularia snails were collected on July 23, 2017 from a pond in Tongliao, Inner Mongolia, China (43°38′2.184″N; 122°15′43.9632″E). The depth of the pond was approximately 0.5 m. After transport to the laboratory, all snails were measured and selected for grouping based on developmental stage. Snails with 9.7 ± 0.5 mm shell height were preliminarily classified as adults and pooled into the adult snails (AS) group. Snails with 5.5 ± 0.4 mm shell height were preliminarily classified as juvenile snails (JS) and pooled into the JS group. Then, the snails were further selected by gonad development: JS with small and thin gonad; AS with full gonads and intumescent, transparent egg mass (Vasileva, 2012; Dikkeboom et al., 1985; Takacs-Vesbach et al., 2016).

Snails were first washed with tap water and then washed with sterile water. A total of 70% ethanol was used to wipe the snail shells. Then, the snails were anaesthetized with MS-222 (Sigma, St. Louis, MO, USA) and all dissections were performed aseptically, using sterile instruments. Part of the marginal shells were carefully broken, and removed to expose the soft body, and then, the whole soft body was removed from shells and washed by sterile water. The digestive tract was carefully isolated from the body, and then, the portion of gut was collected from the stomach (excluding the stomach) to the anus of the digestive tract (Fig. S1). Meanwhile the snails were further selected according to their gonad development. After sampling and classifying, we ended up with four snails from each group. The total time required for collection and dissection did not exceed two hours. The gut and its contents were carefully collected into plastic cryo-tubes, flash frozen in liquid nitrogen, and stored at −80 °C until further analysis.

DNA extraction and PCR amplification

We extracted the genomic DNA using the FastDNA® Spin Kit for Soil (MP Biomedical, Solon, OH, USA) according to the manufacturer’s instructions. DNA yield and quality are shown in Table 1. After size measure, dissection, we have six samples for each of the AS/JS groupings. Two of each stage were filtered after DNA extraction due to low yield, so we finally get four samples of each stage for sequencing. The 338F/806R primer set, targeting the V3–4 region of the bacterial 16S rRNA gene, was used for PCR amplification as described in Dennis et al., (2013). A total of 12-bp barcodes were designed on primers to recognize the sequences of different samples. PCR amplification was performed using the TransStart® FastPfu system (Transgen Biotech, Beijing, China) (Ma et al., 2014). The following PCR cycle conditions were used: one cycle of 95 °C for 3 min, 27 cycles of 95 °C for 30 s, 55 °C for 30 s, and 72 °C for 45 s, and a final extension at 72 °C for 10 min. Each 20 μL reaction mixture consisted of 4.0 μL of 5×FastPfu Buffer, 2 μL of deoxynucleoside triphosphate mix (2.5 mM each), 0.4 μL of FastPfu DNA Polymerase, 10 ng of template DNA, 0.8 μL of Forward Primer 338F (5 μM), 0.8 μL of Forward Primer 806R (5 μM), 0.2 μL of BSA, then, the remaining volume was filled to 20 μL using double-distilled water.

Table 1 DNA yield and quality of the snail bacteria.

Samples	Concentration (ng/μL)	OD260/280	OD260/230	
AS_1	78.10	1.96	0.36	
AS_2	81.20	1.99	0.14	
AS_3	69.40	2.08	0.10	
AS_4	320.10	1.92	0.68	
JS_1	92.90	1.96	0.43	
JS_2	300.50	1.97	0.44	
JS_3	81.80	1.93	0.77	
JS_4	207.90	1.96	0.35	

Illumina amplicon sequencing

PCR products were purified using a Trans PCR Purifcation kit and quantified using the QuantiFluor™-ST System. Each sample was mixed in equimolar amounts. Then sequence libraries were prepared using the NEB Next® Ultra™ DNA Library Prep Kit for Illumina (New England Biolabs Inc., Ipswich, MA, USA) according to the manufacturer’s instructions. The library quality was assessed by spectrophotometry and 300 bp paired-end sequences were generated on an Illumina Miseq platform PE300 (Illumina Corporation, San Diego, CA, USA) with the 600-cycle MiSeq Reagent Kit v3 (Illumina, San Diego, CA, USA) at the Shangai Majorbio Bio-Pharm Technology Co., Ltd. (Shanghai, China).

Sequence data of all samples were deposited in the NCBI Sequence Read Archive under the BioProject number PRJNA438016.

Data analysis

The paired-end sequences were merged into a single sequence with a length of 434 bp using FLASH (Magoc & Salzberg, 2011). The QIIME (version 1.17) pipeline was used to eliminate low quality sequences (i.e., those with >6 bp of homopolymers, primer mismatches, or mean quality score lower than 25) (Caporaso et al., 2010). Chimeric sequences were removed via HCHIME (Edgar et al., 2011). Then, operational taxonomic units (OTUs) were clustered using Usearch (version 7.1; http://drive5.com/uparse/) (Edgar et al., 2011) at a 97% identity threshold. The number of sequences per sample ranged from 27,650 to 37,290. We rarefied each sample to 26,823 sequences, and 100 iterations of the Usearch rarefaction did not quantitatively change results. We used one of the rarefactions at random among 100 iterations to generate OTUs to represent a table that included the resulting 214,584 sequences to be used in all subsequent analyses. Alpha diversity was analyzed via indices of community diversity (Shannon and Simpson) and community richness (Ace, Chao, and Sobs) using mothur software (http://www.mothur.org/) (Schloss et al., 2009). Phylogenetic affiliations of representative sequences were analyzed via RDP Classifier against the SILVA (SSU115) 16S rRNA database with a confidence threshold of 70% (Quast et al., 2013). We used principal coordinates analysis (PcoA) (Lozupone & Knight, 2005) to calculate beta diversity, and subsequently used ANOSIM to confirm findings from the distance matrices.

To identify statistically significant taxonomic groups that differ between JS and AS, we used Welch’s t-tests (confidence interval method: Welch’s inverted, p < 0.05) to compare differences in species abundance between the two groups using the Software of the Statistical Analysis of Metagenomic Profiles (Parks & Beiko, 2010). We also used the linear discriminant analysis effect size (LEfSe) to identify significant associations between bacterial taxa and host groups (JS and AS) (Segata et al., 2011). Metagenomic functional composition was predicted from the latest Kyoto Encyclopedia of Genes and Genomes (KEGG) database (Kanehisa et al., 2012) using the Phylogenetic Investigation of Communities by Reconstruction of Unobserved States (PICRUSt) approach (Langille et al., 2013).

Results

Bacterial complexity of the snail gut flora

A total of 576,400 raw reads were generated using the Illumina Miseq sequence platform and 251,072 high quality sequences were obtained (following quality control and sequence filtration). The mean (± standard deviation) number of sequences per sample was 31,384 ± 4,292 (Table S1) with an average length of 434 ± 1.5 bp. The 214,584 rarefied sequences were clustered into 1,196 OTUs (mean number per sample: 890.75 ± 43.80), with 1,130 and 1,125 OTUs in JS and AS, respectively. The representative sequences for all OTUs are available in Data S1. Ace, Chao, Shannon, Simpson, and Sobs indices indicate no significant differences in the diversity between JS and AS populations (p > 0.05, student’s t-test) (Table 2). The plateau status of the rarefaction curves indicated sufficient depth of sequencing (Fig. S2).

Table 2 Alpha-diversity of the bacterial communities in R. auricularia.

	Ace	Chao	Shannon	Simpson	Sobs	
Juvenile snails	997.78 ± 36.38	1009.20 ± 44.63	5.00 ± 0.22	0.024 ± 0.009	901.75 ± 25.9	
Adult snails	972.78 ± 39.27	974.77 ± 48.68	5.12 ± 0.22	0.019 ± 0.007	880 ± 59.364	
p-Value	0.67	0.47	0.47	0.48	0.67	

Taxonomic composition of gut bacterial community

We characterized the gut bacterial communities of snails. The OTUs that could not be assigned to a specific genus are displayed using the highest taxonomic level that could be assigned (order, class, or phyla).

A total of 10 phyla accounted for 98.9% of the total sequences (Fig. 1A). Proteobacteria (JS: 36.0%, AS: 31.6%) and Cyanobacteria (JS: 16.3%, AS: 19.5%) were the most dominant bacterial phyla, followed by Chloroflexi (JS: 9.7%, AS: 13.1%), Firmicutes (JS: 14.4%, AS: 6.7%), and Actinobacteria (JS: 8.1%, AS: 12.6%). Other phyla with lower abundance were Tenericutes (JS: 7.3%, AS: 6.2%), Bacteroidetes (JS: 3.4%, AS: 2.0%), Fusobacteria (JS: 1.3%, AS: 1.2%), and Verrucomicrobia (JS: 0.7%, AS: 1.6%). One phylum (JS: 1.9%, AS: 4.6%) was not classified. Proteobacteria contained the largest number of OTUs (454), which belonged to the following classes: alpha-, gamma-, beta-, delta-, and epsilon-proteobacteria (Fig. S3), followed by Firmicutes (168), Cyanobacteria (149), Actinobacteria (122), Bacteroidetes (87), and Chloroflexi (70). A particularly high abundance of Cyanobacteria was found in the gut of snails, which may have originated from the snail’s diet.

Figure 1 Relative abundance of bacterial communities in R. auricularia samples.

(A) Phylum level, all remaining taxa with abundance <1% are summarized as other. (B) Family level (or the nearest identifiable phylogenetic level), all remaining taxa with abundance <5% are summarized as other. (C) Genus level (or the nearest identifiable phylogenetic level), all remaining taxa with abundance <5% are summarized as other.

There were 53 identifiable bacterial families with >1% abundance in at least one of the samples (Fig. 1B). Among them, FamilyI_o__SubsectionIII (c__Cyanobacteria), Rhodobacteraceae, Chloroflexaceae, Mycoplasmataceae, Chromatiaceae, FamilyII_o__SubsectionII (c__Cyanobacteria), Cyanobacteria, Lachnospiraceae, Ruminococcaceae, Caldilineaceae, Nocardioidaceae, Acetobacteraceae, Leptotrichiaceae, and MNG7 were the most common families.

There were 54 genera with >1% abundance in at least one of the samples and the sequences of these genera constituted 55.6% of the total number of all sequences. Of all 54 genera, 36 genera were identifiable (Fig. 1C). The 15 most abundant classified genera were Paracoccus, Pleurocapsa, Microcoleus, Thiodictyon, Leptolyngbya, Eubacterium, Subdoligranulum, Nocardioides, Pseudomonas, Faecalibacterium, Chroococcidiopsism, Kluyvera, Rhodobacter, Lemprocystis, and Gemmobacter, with abundances ranging from 1% to 9.9%.

Microbial community analysis

Principal coordinate analysis was used to determine the similarities of gut microbial communities between JS and AS. Unweighted UniFrac distance PcoA showed that JS samples formed a distinct cluster and could be separated from adult snail samples (ANOSIM: Unweighted unifrac, p-value = 0.021, R-value = 0.677). In contrast, when we used weighted UniFrac to account for the abundance information, the samples did not apparently cluster into two groups (ANOSIM: Weighted unifrac, p-value = 0.199, R-value = 0.198) (Fig. 2).

Figure 2 Unweighted uniFrac principal coordinate analysis of the snail bacterial communities.

The juvenile snails are shown by J1, J2, J3, J4, adult snails are shown by A1, A2, A3, A4. ANOSIM: p-value = 0.021, R-value = 0.677.

We also assessed differences in species abundance between JS and AS populations. We found no differences in the abundance of the vast majority of bacteria at both phyla and genera levels (Figs. 3A and 3B). LEfSe analysis (threshold: 4.0) showed that two genera of bacteria were significantly associated with JS, Ruminococcaceae (JS: 5.8%; AS: 1.4%), and Subdoligranulum (JS: 3%; AS: 0.6%) (Fig. 3C).

Figure 3 Taxonomic difference between juvenile and adult groups.

(A) Wilcoxon rank-sum test bar plot of bacterial phyla. (B) Wilcoxon rank-sum test bar plot of bacterial core genera. (C) Diagram of significant associations between gut bacterial taxa and snail population (linear discrimination algorithm LEFSe, Threshold = 4.0). AS represented adult snails and JS represented juvenile snails.

Bacterial community differences and similarities

Venn analyses found that 1,060 OTUs (88.7% of 1,196 OTUs identified) were shared between JS and AS. In fact, 70 unique OTUs were found in JS and 65 were found in AS (Fig. S4). We found 469 core OTUs in all snail samples representing 88.4% of all OTU sequences (Table S2). Among these, 15 core OTUs had a mean abundance >1%, and supplied 33.9% of all OTU sequences. The most abundant core bacterial genera were Mycoplasmataceae, Chloroflexaceae, Paracoccus, Microcoleus, Pleurocapsa, Thiodictyon, Caldilineaceae, leptolyngbya, Eubacterium, Subdoligranulum, and Nocardioides (Table S2).

Functional predictions of bacterial communities

The predicted genomic functions of R. auricularia bacterial community were performed using PICRUSt. The level 1 KEGG pathways indicated a high abundance of predicted functions related to metabolic pathways, environmental information processing, and genetic information processing. The relative abundance of metabolic pathways accounted for 50.8% (Fig. S5).

The level 2 KEGG pathway data (Fig. 4) indicated that the pathways related to membrane transport, amino acid metabolism, carbohydrate metabolism, and xenobiotic biodegradation and metabolism were enriched in both JS and AS samples, with average abundances of 12.4%, 10.5%, 10.0%, and 5.7%, respectively. Further examination of the carbohydrate metabolism pathways indicated an abundance (6.7%) of pathways related to both the starch and sucrose metabolism, including functions for glycoside hydrolysis such as cellulose degradation (Fig. S6). The KEGG pathways of energy metabolism and cell motility, and transcription showed significant differences between JS and AS groups (p < 0.05). The pathways related to human diseases (e.g., infectious and neurodegenerative diseases) were found to have low abundances.

Figure 4 PICRUSt predictions of the functional composition of snails microbiome.

(A) represents KEGG pathway at level 1, (B) represents KEGG pathway at level 2, and (C) represents the abundance of each function pathway. Study sites: JS, juvenile snails; AS, adult snails.

Genetic pathways associated with xenobiotic biodegradation and metabolism maybe play a role in environmental cleaning or bioremediation in the ecosystem. Results showed that the intestinal microbiota were enriched with functions that were related to organic contaminant metabolism (Fig. S7). This included contaminants typically metabolized by the cytochrome p450 family, including benzoate, toluene, aminobenzoate, naphthalene, polycyclic aromatic hydrocarbons, and other similar xenobiotics. Furthermore, some pathways that are typically associated with the degradation of highly toxic matter were also present in high abundance, including those associated with the degradation of dioxins, atrazine, xylene, bisphenol A, and ethylbenzene. This indicated that the gut bacteria of snails may help to degrade anthropogenic pollutants, which could otherwise be harmful to animals and humans.

Discussion

Radix auricularia is a freshwater herbivorous snail of great environmental and ecological importance (AL-Sultan, 2017; Eckblad, 1976). In this study, snails were sampled during the summer, at a time when the snails typically undergo rapid growth due to suitable temperatures and abundant food supply (Guo et al., 2016; Zhang et al., 2018). To compare gut microbial communities at different growth stages, adult and JS were captured. To limit differences of environmental conditions, all snails were sampled from the same aquatic area.

We characterized the gut bacterial community of the snail R. auricularia using next generation sequencing technology. The alpha- diversity indices indicate a high diversity of the R. auricularia bacterial community. The high proportion of shared OTUs and similarities between the most abundant bacterial taxa indicates that adult and JS likely have similar gut bacterial community structures. Using unweighted PCoA plot, samples were clustered according to their growth stage (JS and AS clusters), indicating that the developmental stage may have an effect on gut bacterial community. In contrast, as weighted PCoA showed, despite its high R values, the clustering was not significant (p > 0.05). This is at least in part due to abundance information, which can obscure significant patterns of variation in the taxa that are present (Lozupone et al., 2007; Chang, Luan & Sun, 2011), indicating that taking the abundance of bacterial taxon into account revealed similarities between JS and AS populations that were not detected solely by an examination of phylogenetic lineages. The obtained OTUs (1,196) belonged to more than 10 phyla (predominantly Proteobacteria followed by Cyanobacteria, Chloroflexi, and Firmicutes). At the phylum level, Proteobacteria was identified as the dominant bacteria in R. auricularia bacterial community. Previously, Proteobacteria was also observed as most dominant bacteria in other snails, such as Achatina fulica (Pawar et al., 2012), Helix pomatia (Nicolai et al., 2015), Biomphalaria pfeifferi, Bulinus africanus, and Helisoma duryi (Van Horn et al., 2012). However, at both the family and genus levels, there were differences between our study and the results reported previously, that is, the dominant bacteria for H. pomatia were pseudomonadaceae, enterobacteriaceae, and pantoea, for A. fulica, these were Citrobacter and Enterobacter. In contrast, our results indicated Rhodobacteraceae, Chloroflexaceae, Mycoplasmataceae, Paracoccus, Thiodictyon, and Eubacterium as the most abundant gut bacteria. The differences of bacterial communities between these snails may be caused by species, habitat, physiological states, and environmental changes (Nicolai et al., 2015). The bacterial taxa present in our study (e.g., Pseudomonas, Clostridiaceae, Lactococcus, Bacteroides, Flavobacteriaceae, Mucilaginibacter, Citrobacter, Aeromonas, Acinetobacter, and Sulforospirillum) were also previously reported in the gut of A. fulica (Cardoso et al., 2012b), which demonstrated the occurrence of herbivore and plant-associated bacteria.

Cyanobacteria are widespread throughout aquatic areas and are the main source of energy for snails (Qiao et al., 2018). Cyanobacteria were the second most dominant bacterial taxa in our study: 147 OTUs were assigned to Cyanobacteria, which represented 12.3% of the total OTUs and 17.9% of the total abundance. The second commonly detected OTU (OTU107) was Chloroflexaceae, which is considered to be photosynthetic bacteria (Gupta, 2013). Similar to other herbivores, the high abundance of Cyanobacteria was likely a result of incomplete digestion of exogenous plants (Ye et al., 2014). Among this phylum, Family I (order Subsection III) was most dominant among all measured snail samples (Fig. 1B), suggesting that Family I (order Subsection III) may be an important dietary resource for R. auricularia. Many of the Cyanobacteria bacteria found in the snail gut, including Leptolyngbya, Nostoc, and Pleurocapsa, Microcoleus, Gemmobacter, Exiguobacterium, and Rubrobacter, are of environmental origin, such as fresh water and soil (Hagemann et al., 2015; Lv et al., 2017; Strahsburger et al., 2018; Albuquerque et al., 2014).

As the largest biomass on earth, cellulose and hemicellulose have the greatest potential for the production of biofuels via hydrolytic processes (Lynd et al., 2008). Gut bacterial communities play an important role in the digestion of cell walls and plant lignocelluloses because of the presence of glycoside hydrolases (Morrison et al., 2009). Many bacteria found in our study, such as Paracoccus, Pseudomonas, Aeromonas, Stenotrophomonas, Citrobacter, Bacillus, Micrococcus, Devosia, Shinella, and Rhizobium, have previously been identified as cellulolytic species, associated with carboxymethyl cellulase (CMCase) activity or avicelase activity (Huang, Sheng & Zhang, 2012; Saha et al., 2006; Pawar et al., 2015). Paracoccus, Pseudomonas, and Aeromonas were predominant bacteria in R. auricularia, indicating that they might be important for the cellulose degradation process. Huang, Sheng & Zhang (2012) reported that 70% of the isolated cellulolytic bacteria from the gut of Holotrichia parallela larvae were Proteobacteria, and some of the cellulolytic bacteria belonged to Actinobacteria, Firmicutes, and Bacteroidetes, which is similar to the findings of our study. The genera Klebsiella and Enterobacter were found in the A. fulica gut at a dominant position among the cellulolytic bacterial community; however, they were not found in our study (Pawar et al., 2015). Paracoccus, may not only be important cellulolytic bacteria as described above, but also have been found to be a potential bacteria for bioremediation of PAHs-contaminated soils (Teng et al., 2010). Although many members of Pseudomonas are animal and plant pathogens, some members of the genus are able to degrade chemical pollutants in the environment, such as polycyclic aromatic hydrocarbons, and carbon tetrachloride (O’Mahony et al., 2006; Sepúlveda-Torres et al., 1999). As a member of the Enterobacteriaceae family, Aeromonas inhabit fresh and brackish water and are responsible for human intestinal diseases (Parker & Shaw, 2011). However, in snails, Aeromonas are one of the cellulolytic bacteria. In summary, in this study the cellulolytic bacteria found in R. auricularia are not only centrally important due to their role in the degradation of cellulose and other plant wall components, but they are also important due to their role in bioremediation of the ecosystem.

Our results show that the most abundant OTU (OTU585) were affiliated with Mycoplasmataceae, which belongs to the phylum Tenericute. Mycoplasma has been implicated as an infectious species that can colonize humans and a wide range of animal species, causing diseases in the hosts (Biondi et al., 2014). The predicted functions of infectious and human diseases that were identified among the KEGG pathways could potentially be associated with the genus Mycoplasma.

Although previous research has confirmed that bacterial communities vary during host development and growth (from birth to adulthood) (Nistal et al., 2012; Stephens et al., 2016), other studies have shown that the bacteria communities are relatively similar between juvenile and adult stages in a variety of animal hosts (Xue et al., 2015; Hird et al., 2014). Our study also showed that taking the abundance of bacterial taxon into account (weighted UniFrac) revealed similarity of bacterial communities between JS and AS populations. The enrichment of Faecalibacterium and Subdoligranulum (both belong to Ruminococcaceae) in JS and their poor presence in AS is commonly found in many other animals (Gu et al., 2013; Dethlefsen & Relman, 2011). These bacteria have been found to be highly beneficial to their hosts, by producing butyrate and other short-chain fatty acids via fermentation of dietary fibers (Miquel et al., 2013; Flint et al., 2012). These biomarkers may be important for JS in terms of improvement of digestive ability, boosting their immune system, and other similar physiological functions (Gu et al., 2013; Flint et al., 2012).

To understand the role of gut bacterial community in snails, we explored the function of gut bacteria using PICRUSt (based on the 16rRNA gene data). The obtained results indicated that the microbiome taxa are related to many physiological functions, which may aid their hosts (Sommer & Bäckhed, 2013). Previous studies with the snail A. fulica showed that many particular functional genes in the gut microbiota (e.g., genes associated with the production of amino acids, fatty acids, cofactors, vitamins, and enzymes) are required by the hosts for plant fiber degradation (Cardoso et al., 2012a). As recently reported by Joynson et al. (2017), 2,510 genes corresponding to glycoside hydrolase activity and 561 carbohydrate-binding modules were identified in a total of 108,691 putative genes of the gut microbiome of the common black slug Arion ater. The microbiotic function predicted in our study are also necessary for many physiological functions. In fact, the richness of cellulolytic bacterial taxa could lead to the isolation of bacterial cellulases from snails (Pinheiro et al., 2015). Furthermore, the discovery of bacteria related to xenobiotic biodegradation illustrates the role of snails in the degradation of environmental contaminants, indicating the potential application of the snail microbiota for environmental cleaning, which was also found in other animal or environmental microbiota (Yang et al., 2015; Zhou et al., 2016).

Conclusions

The use of advanced molecular technology offers a new method to study microbial communities based on their DNA. In this study, we used the high-throughput sequencing technique to investigate the bacterial diversity of individuals of the snail R. auricularia and predicted metagenomic functions using PICRUSt. This work demonstrates that the phyla Proteobacteria, Cyanobacteria, Chloroflexi, and Firmicutes were predominant in the microbial community. A high number of OTU and genus diversity were shown. Growth and gonad development may have influenced the taxonomic characteristics of the gut bacterial community without influencing the predicted function of gut bacteria. For R. auricularia, the potential for isolating cellulolytic bacteria and environmental cleaning are indicated by the abundant presence of cellulolytic bacteria and metagenomic functional predictions. Further research is required to better characterize the interaction between gut flora and their hosts in snails such as R. auricularia.

Supplemental Information

Supplemental Information 1 Table S1. The sequence data of all snail samples.

A_1, A_2, A_3, A_4 represented adult snail samples, J_1, J_2, J_3, J_4 represented juvenile snails samples.

Click here for additional data file.

Supplemental Information 2 Table S2. Taxa of core bacteria shared by all R. auricularia samples and their relative abundance.

The core bacteria with abundance >0.5%.

Click here for additional data file.

Supplemental Information 3 Data S1. OTUs represent sequence data.

Click here for additional data file.

Supplemental Information 4 Fig. S1. Digestive system of the snail R. auricularia.

Photo credit: Zongfu Hu.

Click here for additional data file.

Supplemental Information 5 Fig. S2. Rarefaction analysis of observed richness of the snail bacterial communities.

There were eight samples collected and successfully sequenced. Four samples were collected from juvenile snails and four samples from adult snails. Operational taxonomic units (defined at 97% sequence similarity) identified by Illumina Miseq sequencing of the V3-4 region of the 16S rRNA genes.

Click here for additional data file.

Supplemental Information 6 Fig. S3. The sequence proportion of class taxon in Proteobacteria.

Click here for additional data file.

Supplemental Information 7 Fig. S4. Venn analysis of shared OTUs and unique OTUs between adult snail (AS) and juvenile snail (JS).

Click here for additional data file.

Supplemental Information 8 Fig. S5. The abundance of genes related to predicted function at KEGG level 1.

Click here for additional data file.

Supplemental Information 9 Fig. S6. The abundance of genes in the subsystem carbohydrate metabolism.

Click here for additional data file.

Supplemental Information 10 Fig. S7. The abundance of genes in the subsystem Xenobiotics biodegradation and metabolism.

Click here for additional data file.

Additional Information and Declarations

Competing Interests

Author Contributions

Field Study Permissions

Data Availability

The authors declare that they have no competing interests.

Zongfu Hu conceived and designed the experiments, performed the experiments, analyzed the data, contributed reagents/materials/analysis tools, prepared figures and/or tables, authored or reviewed drafts of the paper, approved the final draft.

Xi Chen performed the experiments, authored or reviewed drafts of the paper, approved the final draft.

Jie Chang performed the experiments, authored or reviewed drafts of the paper, approved the final draft.

Jianhua Yu performed the experiments, contributed reagents/materials/analysis tools, authored or reviewed drafts of the paper, approved the final draft.

Qing Tong analyzed the data, prepared figures and/or tables, authored or reviewed drafts of the paper, approved the final draft.

Shuguo Li performed the experiments, authored or reviewed drafts of the paper, approved the final draft.

Huaxin Niu conceived and designed the experiments, analyzed the data, contributed reagents/materials/analysis tools, authored or reviewed drafts of the paper, approved the final draft.

The following information was supplied relating to field study approvals (i.e., approving body and any reference numbers):

Research permits were provided by the Forestry Bureau in Tongliao (TL218) and Inner Mongolia University for Nationalities’s Institutional Animal Use and Care Committee (2016-IMUN-029).

The following information was supplied regarding data availability:

The sequence data are supplied as a Supplemental File.

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
