# Peer review of "Compositional and predicted functional analysis of the gut microbiota of Radix auricularia (Linnaeus) via high-throughput Illumina sequencing"

_PeerJ, doi:10.7717/peerj.5537_

## Round 0.1 · original submission · Major Revisions

Three detailed and reasoned reviews have been received. The consensus was that this work will require major revision before further consideration. Please address their comments specifically, if you choose to resubmit. Importantly, provide a stronger justification for this work. As submitted, Abstract states “..plays and important role (line 34)..” but then launches into methods.

The text needs revision for clarity and methods requires explanation. Address reviewers concerns about animal collection and preparation and PCR-amplification. It would improve the manuscript to provide the DNA yield and purity data. Explain the meaning of results in Abstract. It is not clear if the “...large number of species..not identified..” means that many novel ribotypes were detected or if anything functions of interest were predicted.

Line 64. Phrases like “previous studies have shown that” and “It has been proposed that..(line 68)”, “It has also been found..(line 296), “is believed to an (line 297)”,.do not add any information.
Line 80. The statement “Generally speaking, snails have weak locomotive skills..” may be humorous to some readers.
Line 96. I am not sure how the “environmental cleaning functions” were assessed.
Line 140. Change to SILVA
Line 163. Instead of sending the reader to look at a Figure, state what the Figure shows.
Line 167. Here and throughout use appropriate significant figures (36 not 35.98).
Line 212. As before, what does the data show?
Line 285. Delete pronoun “The”

Follow a consistent format in References. Do not include the issue. only vol:pages. Only cap proper nouns in titles (line 359).

Reviewer 1 ·

Basic reporting

This manuscript examined the gut microbiome of Radix auricularia, in both juvenile and adult stages. I find the topic interesting, although the premise and importance of the study is not clearly described. The introduction should clearly state why is it important to study the microbiome of this snail, and why it is important to study various life stages. The main argument of most paragraphs was unclear. Paying attention to the flow, focus, and structure of each paragraph should improve the clarity of the writing. I recommend the authors focus upon a clear topic sentence for each paragraph, with only relevant supporting sentences in the remainder of the paragraph. For example, I am unclear as to how the four sentences in lines 85-91 are related to one another, and the main point the authors are trying to convey.

The language used in this manuscript is not quite clear or professional. For example, lines 58-59. The object that "this" refers to is unclear, and the sentence should be cited. Phrases such as "a few" (line 178), "fairly similar" (line 266), and "most other" (line 323) should be avoided.

The figures are relevant, although I would consider placing Figure 1 in the supplemental material and only showing on taxonomic classification level in Figure 3.

Other specific comments:
Line 41,43,174- numbers at the beginning of sentences should be spelled out.
Line 42- species or OTUs?
Please fix types in line 52. Also, it is unclear what is meant by "Pulmonata Mollusca".
Line 84- I believe a snail species is supposed to be listed here?

Raw data is not shared.

Experimental design

As discussed above, the importance and purpose of the research questions are not well defined. With regard to sequencing and analysis, the experimental design is sufficient although replicate numbers are quite low (4 individuals per category). However, many more details are needed throughout the Materials and Methods section, particularly for the collection and assessment of the snails. For example, why were 40 individuals selected, but only 8 individuals sequenced? What methods were used to examine gonads? Where is this data? How do the authors know whether snails had aged over winter (line 107). What is meant by "gut" in line 114. Does this include the stomach? What PCR reagents were used, in what quantities, and what were the PCR conditions? What clustering methods were used? Were data rarefied? If not, please justify. Please also be sure the methods for all reported result are described. Not all diversity index calculations are described in the materials and methods. LEfSe analysis is also not described in the M&M. The abbreviations AS and JS were never defined in the text.

The description of the results is also a little ambiguous. For example, in line 208 I'm unclear what "composing 88.70% of all OTUs numbers" means. This applies to lines 205 and 214 as well.

Validity of the findings

I would recommend that the authors focus less upon the exact taxonomic composition and place more emphasis on the significance and ecological implications of the bacteria detected. For example, emphasis is placed on the presence cellulolytic species in Lines 290-293. The authors should discuss the ecological importance of cellulolytic species in the context of their results.

How is Radix auricularia different than those listed in lines 85-87,253-255? Why would their gut microbiota be different than the snails already studied? In lines 243-249, many assumptions are made that were not tested (such as similarity in diet composition). Please either prove these assumptions or remove them.

Additional comments

Assessment of functional genes was not actually carried out, rather predicted functional genes. This is an important distinction. Please change the title to read "… predicted functional analysis…" or something similar.

In line 132 QIIME is cited incorrectly.

·

Basic reporting

The authors describe bacterial communities from the gut of some juvenile and adult R. auricularia individuals. The authors used Illumina technology and 16S rRNA sequencing to determine the taxonomic profiles of these communities. I believe that the data are of interest as very little is known about snail microbial communities. Furthermore, a few other studies have shown that these animals are a promising source of new microbial enzymes with potential technological applications and of ecologic significance. Unfortunately, several language mistakes and other errors such as missing references were left in the text, many of which would have been easily avoided had the authors revised their work more carefully. Those that I could find are listed below:

Introduction:
Line 52: please change to: Radix auricularia (Linnaeus, 1758), a Pulmonata mollusk, is naturally…
Line 59: please change to: …have demonstrated that snail gut bacteria could assist…
Line 60: please change to: …vascular plants…
Line 67: please change to: …fix nitrogen…
Line 106: please change to: …adult snails were over 1 year old and juvenile snails were less than 1 year old.
Methods:
Line 120: Dennis et al (2013) is not on your references. Please correct this.
Line 123: Ma et al (2013) is not on your references list. Please correct this.
Line 128-129: please change to: spectrophotometry and paired-end sequences were generated on a Illumina Miseq platform (Illumina Corporation, San Diego, USA) with the 600-cycle MiSeq Reagent Kit v3 at the Shangai…
Line 132: The reference for Qiime that the authors provided (Lai et al. 2014) is not adequate. They should cite: QIIME allows analysis of high-throughput community sequencing data. J Gregory Caporaso, Justin Kuczynski, Jesse Stombaugh, Kyle Bittinger, Frederic D Bushman, Elizabeth K Costello, Noah Fierer, Antonio Gonzalez Pena, Julia K Goodrich, Jeffrey I Gordon, Gavin A Huttley, Scott T Kelley, Dan Knights, Jeremy E Koenig, Ruth E Ley, Catherine A Lozupone, Daniel McDonald, Brian D Muegge, Meg Pirrung, Jens Reeder, Joel R Sevinsky, Peter J Turnbaugh, William A Walters, Jeremy Widmann, Tanya Yatsunenko, Jesse Zaneveld and Rob Knight; Nature Methods, 2010; doi:10.1038/nmeth.f.303
Line 135: please change to: Using Usearch (version ….

Results:
Line 159: please change to: …between juvenile and adult populations.
Line 163 please change to: …are illustrated in Figure 2.
Line 184. This is an awkward sentence. I am not sure what it means. Please explain and rewrite it more clearly.
Line 197: please change to: …at both phylum and genus levels, there were no differences…
Line 219: please change to …KEGG pathways…
Line 225-226: Please change to: Figure S5 shows, according to KEGG mapping using the KO system, genetic pathways associated with xenobiotic biodegradation and metabolism.

Discussion:
Line 244: please remove the sentence: Furthermore…..similar. As this essentially says the same as the previous sentence.
Line 249: Please introduce a paragraph for: In this study.
Line 251: please remove the word taxa after phyla.
Line 256: Cardoso et al. 2012 did not study H. pomatia. They studied Achatina fulica.
Line 257: Please change to: Bacteroidetes
Line 282: Please change to: Pleurocapsa.
Line 286: Please change to: …lignocelluloses because of the presence of glycosyl hydrolases.
Line 287: Please change to: Many bacteria…
Line 294: Please change to: …most abundant OTU (OTU585) was affiliated with…
Line 309: Please change to: …account…
Line 312: Please change to: These bacteria….
Line 319: bacterial hosts is awkward.
Line 325: Please change to: …isolation of bacterial cellulases from snails….

Conclusion
Line 330: Please change to: …of individuals of the snail R. auricularia was described…
Line 332: Please change to: A high OTU and genus diversity were shown.
Line 333: Please change to: Growth and gonadal development may have influenced…

Suggestions to improve the Table and Figures:
Table and figures
Table 1- Please remove the text altogether and just leave the title of the table. You already mentioned in the manuscript text that there no statistically significant differences between the alpha diversity metrics and which text was used. I would also remove Sample/Estimators from the table as the headings are self-explanatory.
Figure 1- In the title please change to: …snail bacterial communities.
Figure 1- In the legend please change to: Four samples were collected from juvenile …
Figure 1- In the legend change to: sequencing of the V3-V4 region of the 16S rRNA gene.
Figure 2- In the legend change to: …taxa with abundance <0.01% are…
Figure 2- I suggest using only major taxa in the figure as there are too many taxonomic ids which makes the figure quite polluted. I would keep 1% for the phylum but increase to at least 5% for the family level and 10% for the genus level. You can give the taxonomy for every OTU and a full taxonomy table on supplementary files.
Figure 5- In the title please change to: PICRUSt predicts the functional composition of snail microbiomes.
Figure 5- In the legend please change to: represents instead of represent.

The references are sufficient but as outlined above some are missing in the reference list.
The authors supply a list of the representative sequences for some of the OTUs. However, they do not supply an accession number for their raw sequence data nor mention if they were submitted to a public database, This is not acceptable.

Experimental design

The work is within the aims and scope of PeerJ and the research questions are on the whole well defined, relevant and meaningful. As stated above there is still not enough data on snail gut microbiota and microbiomes even though these animals are a potential source of novel bacteria and enzymatic activities. Although the data are significant it would have been better if the authors had analyzed at least 10 animals from which group as groups with only 4 animals really make sound statistical analysis more difficult. But, as I said before, due to the scarcity of data on snail gut bacterial communities, especially when one considers the high biodiversity of this group, I still believe that data are useful as a preliminary report. I have some comments to improve the manuscript outlined below:
Methods:
Line 110-111: Please provide a drawing or photograph depicting the different parts of the snail gastrointestinal tract and indicate which parts were used for DNA extraction.
Line 126: The authors state that a library prep kit was used for making sequencing libraries but this strikes me as odd as amplicons under 500-600 bp are usually directly sequenced. The authors say that their amplicons had barcodes so I see no utility in using a library prep kit. The authors should clarify this.
Results
Line 186- What is this high abundance? More than what percentage?
Lines 191 -195- PcoA analysis- In contrast to what the authors state there are differences in community structure between juvenile and adult snail on both unweighted and weighted unifrac distances from what the graph shows. I suggest that a statistical analysis such as adonis, ANOSIM, BIO-ENV, Moran’s I, MRPP, PERMANOVA, PERMDISP, or db-RDA be used to on the distance matrices to confirm this observation.
Line 199-200: please indicate the abundance (%) of these genera in relation to the total number of genera in juveniles and adults.
Line 216- PICRUSt does not estimate the number of sequences but estimates the number of proteins within pathways estimated from genomes from taxonomically related bacteria, so remove sequences. I suggest using genes instead of sequences. The same goes for line 220 where the authors say : sequences related to human…
Discussion
Line 244: please change to: …so diet was probably similar between sample groups.
As the authors did not show results for gut contents they cannot assume that the diet was highly similar. They can only suppose it was.

Validity of the findings

No comments as most of these points have been addressed in the previous sections.

Additional comments

All of the relevant comments are in sections 1 and 2.

Reviewer 3 ·

Basic reporting

The overall structure of the manuscript is sound, and the figures and data reporting is acceptable. The authors should consider the following:

1. The internal structure of the manuscript makes it extremely difficult to follow. Each method, software program, experimental step, etc. needs to be clearly stated; the "how" and "why" are often confounded or absent. The authors must make their narrative more accessible to the reader by following a consistent mention of the methods both in terms of the order of presentation and in completeness. As a reviewer I have enough experience with what the authors are doing to mentally fill in the gaps as I read, but readers of the manuscript should not be expected to do so. A specific example: the authors used STAMP (the figures are easily recognizable) but there is no mention of it.

2. Many method and literature sources are explicitly or implicitly referenced throughout the document are not referenced and/or listed in the Literature Cited. The literature that is cited is not in alphabetical order.

3. The authors should have any future re-submissions checked by a native English speaker to improve basic grammar and punctuation.

4. I would recommend that the authors other relevant and/or recent published snail gut papers, including Joynson et al. (2017), Aronson et al. (2016), Van Horn et al. (2012) to name a few. These either represent the most recent publications or similar analyses from other freshwater snails.

5. Figure 2 needs to be reworked for easier interpretation, as does Figure 4.

Experimental design

The general experimental design is acceptable and consistent with other published works on snail gut microbe studies. My concerns with the experimental design are:

1. The authors crushed the snails to remove the tissue for analysis. This poses a few issues. First, Radix are large enough that when anesthetized they can be removed from the shell intact and the needed parts dissected. Second, although the shells may have been cleaned, snail shells harbor bacterial biofilms on the surfacce and in the fluid trapped in and around the body inside. By crushing the shell, the authors risked contaminating the gut flora with non-gut flora.

2. The sample size the authors used is very small, which is evident in their rarefaction curves.

3. In reporting the microbial results, the authors make no mention of separating the identified bacteria by possible origin. For example, cyanobacteria are most likely dietary and not enteric. The same holds for functional results; infectious and human diseases are identifiable KEGG pathways but are they biologically relevant to the analysis?

4. The 'standard' LEfSe cutoff for reporting is +/- 2 on the log LDA score. The authors used 3.5. Why?

Validity of the findings

I find the findings to be acceptable, the data properly generated and analyzed, and the conclusions to be supported by the results.

Additional comments

This is a sound study in theory, but isn't anywhere near ready for publication as it stands currently. The version sent in for review reads and looks like a very early draft of a manuscript that needs a lot of revision and improvement. I think there is value in the project's conception and execution, but the text belies this fact. I think with major revisions and extensive editing, the authors can have a quality publication on their hands, but not at this point.

---

## Round 0.2 · Major Revisions

We received two reviews of the revised manuscript. The new reviewer called for major revisions. I have also provided comments. Several of these edits are repeated from the first submission, as they were not specifically addressed in the resubmission. If you choose to resubmit, include a rebuttal letter that details your response to each comment below and those detailed by the reviewer.

Line 53. Delete “The” and throughout use this article appropriately.
Line 66. Replace “Their” with “The”
Line 67. Replace “aids their …by utilizing” with “increases their ability to utilize a variety””
Line 71. Delete “The”
Line 193 and 258. As suggested previously, start each paragraph with a topic sentence. Instead of sending the reader to look at a Figure, state what the Figure shows.

·

Basic reporting

no comment

Experimental design

no comment

Validity of the findings

no comment

Additional comments

I am satisfied with the modifications undertaken by the authors and I think the paper is now suitable for publication.

Reviewer 4 ·

Basic reporting

This study seeks to evaluate the microbiome composition and diversity of juvenile and adult Radix auricularia snails and to predict the effect that the gut bacteria may have on host functions. The study is very interesting, is generally well done, and helps fill an important gap in our knowledge. For being the second most specious phylum on earth, mollusks receive very little attention, and their microbiomes are relatively uncharacterized. This study provides important new data and insights in this regard.

There are still quite a few English language problems throughout the manuscript and I think that the manuscript would benefit from another round of editing from a native/proficient English speaker. There are also some instances where I feel that additional citations are necessary. The figures in general look great, there are a few edits that need to be made to some legends (detailed below), and the font in Figure 5 does not match the rest of the figures.

Experimental design

This article is in line with the aims and scope of PeerJ and I feel that this paper meaningfully addresses their questions. I think that this paper addresses an important knowledge gap and authors could provide a greater emphasis on this aspect of their work. In general, I feel that the methods used were rigorous and that results were evaluated accurately.

I only have a couple issues with experimental design. First, the authors need to provide further explanation as to why they chose to only use 4 samples from each developmental stage. Second, the unweighted and weighted analysis give different results. However, this discrepancy is never addressed or discussed in the text, and it needs to be. Otherwise, the methods are explained very well and in a way that would be conducive to replication of the study.

Validity of the findings

The data presented appear to be robust. After the authors address the issue of the weighted vs. unweighted analysis, I think that the results are statistically sound and interpreted in a reasonable manner. The authors generated novel data and interesting new conclusions that increase what we know about snail microbiomes, which thus far have been largely unstudied.

Additional comments

1. Throughout the manuscript the authors use a lot of passive voice. I recommend revising a bit to include more active voice. This will increase readability. e.g. L167-169 could say: “We used principal coordinates analysis (PcoA) to calculate beta diversity, and subsequently used ANOSIM to confirm findings from the distance matrices.”

2. Make sure to use consistent fonts in all figures. Some fonts use serif fonts, others use sans serif fonts. Being consistent in figure formatting with increase aesthetic appeal of the paper.

3. The weighted analysis did not show this same pattern as the unweighted analysis, that analysis found no difference between JS and AS, suggesting no difference in microbiome composition based on developmental stage. Please acknowledge and explain this discrepancy

L21: type in title: should say “microbiota”

L46: the term “data information” is awkward, and not specific enough. Be more specific: e.g. “A large number of species could not be successfully identified, which could indicate the detection of novel ribotypes or result from insufficient availability of snail microbiome data.”

L48-50: this is an awkward sentence and I’m not totally sure what the authors are trying to say. predicted functions of what? What “genes” are the authors referring to if they only did16S sequencing of the microbiome, or are they referring to the snail?

L49: And do you mean “enriched” instead of “rich”?

L53: should say: “searching for xenobiotic…” please revise

L57: change “Pulmonata mollusk” to “pulmonate snail”

L62: harmful to

L66: The capacity

L66-69: the grammar of this sentence it awkward (particularly the term “aids their life”), please revise

L78: “previous”, not “precious”

L84: change “are even stronger impacted by” to “are more strongly influenced by”

L86-L89: or that reproductive mode influences microbiome composition, which way this relationship goes is still unclear

L98: “freshwater”

L102: add a sentence here briefly summarizing what you found and why it’s important.

L111: For grouping based on developmental stage? please specify

L112-115: Is there literature to cite to support these groupings?

L116-117: Please justify/explain why only 4 snails were used for each developmental stage. Of the 40 snails collected, did only 4 meet the criteria for each of the AS/JS groupings?

L121: change to “removed to expose”

L142: “products” not “productions”, and change “through a” to “using a”

L153: change “merged to a” to “merged into a”

L161: how was the single rarefaction chosen? At random?, change represent to generate

L162: change “sequences in all” to “sequence to be used in all”

L 185: The two instances of +/- have different formatting. Is one in bold?

L186: What is the standard deviation for sequence length?

L186-187: awkward sentence, change to something like: “the 214,584 rarefied sequences were clustered into 1,196 OTUs…”

L202-203: include the numbers here

L205-209: these sentences could be more concise, combined into a single sentence, and added to the previous paragraph

L210-211: change “at least one of samples” to “at least one of the samples”

L235: this sentence is unnecessarily complicated, change to something like: “Venn analyses found that 1060 OTUs (88.7% of the 1,196 OTUS identified) were shared between JS and AS.

L244: do the authors mean the predicted genomic functions of the bacteria in the snails’ guts/the snail functional pathways that the bacteria are predicted to influence? This sentence/figure 5 need clarification

L250: should say “enriched” not “rich”

L255: add the word “pathways” after metabolism

L255-257: this sentence could be part of the previous paragraph

L259: change “may be” to “maybe”

L265: do you mean present in high abundance?

L267: change would to could

L277: get rid of “In this study”

L278: “next generation” not “new generation”

L279: add the word “bacterial” before “community”

L281: add the word “community” before “structures”

L282: change to “indicating that the developmental stage may have an effect…”

L282-284: however, your weighted analysis did not show this same pattern, that analysis found no difference, suggesting no difference in microbiome composition based on developmental stage. Please acknowledge and explain this discrepancy

L285-288: this is an awkward sentence, please revise

L289: change different to differences and before to previously

L295: change bacteria to bacterial

L298: bacteria are NOT closely related to snails, I do not think this is what the authors mean to say here. The sentence needs to be revised and clarified

L307-310: please provide citations for this information

L236: change “not only” to “may not only”

L338: change indicated to implicated

L340-341: this sentence is poorly written, please revise to something like: “The predicted functions of infectious and human diseases that were identified among the KEGG pathways could potentially be associated with the genus Mycoplasma.”

L341-343: this could be moved to the section about Cyanobacteria. As they are both potentially dietary, it makes sense to discuss them together.

L347: change “several” to “other”

L348-350: Again, the weight analysis gave a different result. This discrepancy needs to be acknowledged and discussed.

L354-L356: please provide a citation for this sentence.

L357: change expose to understand

L359: Do you mean TAXA, rather than GENES? All of the results are based on the sequence of a single gene (16S). This is unclear and greatly needs to be addressed.

L363-372: Again, I think that you mean the bacterial taxa are associated with these functions, or that there is some evidence that the bacteria affect host genes involved in these processes. Please clarify what you mean by this relationship…because you only sequenced a single gene from a wide diversity of taxa.

L380: high number of OTUs

Figure 1: should go in the supplement

Figure 3: change “were showed”, to “are shown”

Figure 4 legend, change get rid of “taxon” in “bacterial taxon phyla” and “bacterial taxon core genera” and change “bacterial taxon and snail population” to “bacterial taxa and snail population”

Figure 5: change to a sans serif font so that it matches the other figures

---

## Round 0.3 · Minor Revisions

When these minor changes are complete, I think the manuscript will be acceptable for publication.

Reviewer 4 ·

Basic reporting

This new version of the manuscript is much improved. There are still many English language/grammatical errors detail below. Most importantly, the authors still need to address one of the previous reviewer comments regarding genes and their functional analysis. The authors only sequenced 16S and used taxonomic assignments to predict the functional effects of the bacteria themselves and the KEGG pathways that the bacteria are expected to influence. However, in the results section these data are still described as genes that are enriched in the bacteria, which is not what was done. The authors did a great job of addressing the weighted vs. unweighted analysis differences. After the one major issue mentioned above and the minor grammatical errors are addressed I believe that this manuscript will be ready for publication.

Experimental design

None

Validity of the findings

None

Additional comments

L34: change “The gut microbiota is involved” to “The gut microbiota are involved”
L78: change “and previous study showed” to “and previous studies have shown”
L85: change “Gut microbiota is” to “Gut microbiota are,” remove “the” before reproductive, and change “process” to “processes”
L101: change “The results of study here may fill” to “The results of this study fill”
L102: change “of technological” to “for technological”
L113: change “preliminary” to “preliminarily”
L115: change “preliminary” to “preliminarily”
L116: change to “juvenile snails”
L117: change to “adult snails”
L119: change to “Snails were”
L123: change to “washed with sterile water”
L110-L128: In this section the authors need to say how many snails they ended up with after sampling and classifying (4 from each group).
LL160-162: This sentence does not belong in this section; it should be in the “DNA extraction and PCR amplification section.”
L171: change “co-ordinate” to “coordinate”
L173: change “To inspect the statistically significant differences of taxonomic groups between JS and AS” to “To identify statistically significant taxonomic groups that differ between JS and AS”
L217: change “identified” to “identifiable”
L230: change “level” to “levels”
237: specify whether “these OTUs” refers to the 469 core OTUs, or to the 70 JS and 65 AS unique OTUs.
L243: change “pathway” to “pathways”
L248: change “enrich” to “enriched”
L242-264: As noted in the previous round of reviews, “genes” is not the appropriate term here, and is actually misleading. The authors only sequenced 16S from the gut bacteria. The authors should be referring to predicted functions of the bacteria themselves and the PATHWAYS that they might influence, not specific genes. This needs to be changed throughout this entire section, as it is not an accurate representation of what was actually done in this study. The title of this section (16S rRNA gene functional prediction) also needs to be changed, as it is also not an accurate portrayal of what was actually done here. A more accurate section heading would be something like “Functional predictions of bacterial communities.”
L280-281: “partly due…are present”, the grammar in this part of the sentence is awkward. Change to “this is at least in part due to abundance information, which can obscure significant patterns of variation in the taxa that are present.”
L285: change to “At the phylum level”
L286: change “ as the dominant”
L299: get rid of phrase “bacteria related to”
L325: change to “of the cellulolytic”
L340: change “was affiliated” to “were affiliated”
Fig. 4: change “PICRUSt predict the” to “PICRUSt predictions of the.” In the legend change “represent” to “represents” in all occurrences.

---

## Round 0.4 · accepted · Accept

The manuscript may still require some edits for style but we can address those during production.

Michael

#